# The Importance of the Circular Economy Concept among Organizations within the Food Sector and a Management Systems Perspective

**Piotr Kafel** [1,*] and **Krzysztof Rudziński** [2]

1 Department of Quality Management, Krakow University of Economics, 31-510 Kraków, Poland
2 Department of Organizational Behavior, Krakow University of Economics, 31-510 Kraków, Poland; rudzinsk@uek.krakow.pl
* Correspondence: kafelp@uek.krakow.pl

**Abstract:** The concept of the circular economy is becoming an increasingly important issue within the economic and political sphere. The aim of the study is to check the opinions of representatives of organizations on the need to act in accordance with the principles of the circular economy concept in organizations and to indicate the differences in these opinions, taking into account the following aspects: the number of implemented management systems and the FSMS implementation. The survey method was used in the study. The surveyed organizations were divided into groups according to two criteria: having implemented food safety management systems and the total number of implemented management systems. The Mann–Whitney U test was used to analyze differences between groups. The conducted research showed that the organizations with more than three MSSs perceive a greater need to act in accordance with the principles of the CE than other companies. There was no sufficient proof to support the hypothesis concerning the higher engagement of food sector organizations in circular economy activities. The size of the organization is a factor that is also irrelevant in this context.

**Keywords:** circular economy; management systems; food safety; food industry

## 1. Introduction

Currently, one of the main problems in the world is the constant deterioration of the natural environment. This raises concerns about the impact of industrial systems on the resilience of ecological and social systems, i.e., on the ability of ecosystems and society to cope with these changes [1]. The modern linear system of economic functioning has contributed to the acquisition of natural resources, which are then transformed into products (with a short lifespan, designed for one purpose) and utilized in the post-use phase. This pattern dominates both production and consumption and has contributed to generating enormous amounts of waste [2]. Moreover, overproduction to meet the growing needs of the population requires the use of huge amounts of natural resources, which in turn are depleted [3]. The solution to environmental problems resulting from the functioning of the linear economic model is to move towards more circular solutions.

The concept of a circular economy is a relatively new approach to the functioning of the economy. Unlike the linear model, the circular economy concept places particular emphasis on the aspect of product value, which is preserved for as long as possible. This allows one to reduce the amount of waste generated or eliminate it completely [4]. One of the key principles of the circular economy concept is that waste is food. Therefore, all resources, materials and products are reused as input elements in the production of new products [5,6]. However, following the assumptions of the circular economy concept is a difficult task, especially for the lowest economic levels, i.e., organizations, due to the limited number of guidelines of the circular economy concept. Currently, there are two

documents available providing guidelines for implementing the assumptions of the circular economy concept: British BS 8001: 2017 and French X PX 30-901: 2018 [7]. However, there are already well-known tools that allow one to achieve some of the assumptions of the circular economy concept. These are management systems. An important management system standard (MSS) that may be important for the implementation of the principles of the circular economy concept in organizations is an EMS consistent with the requirements of the ISO 14001 standard. The implementation of this system reduces the amount of waste generated [8], and has a positive impact on the organization's results by reducing greenhouse gas emissions [9]. Another example is the implementation of an energy management system (ISO 50001), which aims to reduce pollutants emitted into the atmosphere by organizations, i.e., greenhouse gases [10]. Other management systems can also support the implementation of the circular economy concept, e.g., reducing waste by improving production processes can be implemented using food safety management systems (FSMSs), e.g., compliant with the requirements of the ISO 22000 standard [11]. The aspect of waste is particularly important in the circular economy concept. The food industry wastes a lot of raw materials. For example, the food industry in the UK alone generates 5.2 million tonnes of food surplus, by-products and waste [12]. Food waste also contributes to greenhouse gas emissions. According to a UNEP report [13], it is estimated that around 8 to 10% of total greenhouse gas emissions come from food that is not consumed. Due to the huge impact of organizations on the natural environment, it is important to check their opinion on whether following the principles of the circular economy concept is necessary.

The purpose of this article is to check the opinions of representatives of organizations on the need to act in accordance with the principles of the CE concept in organizations and to indicate the differences in these opinions, taking into account the following aspects: the number of implemented management systems and the FSMS implementation. The conducted research seeks answers to the following research questions:

- Do organizations with implemented management systems see the need to act in principles with the assumptions of the circular economy concept?
- How important are management system standards in recognizing this need?
- Do organizations from the food industry have a higher need to follow the principles of the circular economy concept than other groups of organizations?

The first part of the article presents a literature review. It identifies environmental problems and the resulting need to act in accordance with the assumptions of the circular economy concept, the connections between the circular economy concept and management systems, pointing to common aspects of their implementation, and the connections between the circular economy concept and the food industry. The next part is the empirical one, which defines the research objectives, hypotheses and methods, as well as the results and discussion of the research results.

## 2. Literature Review

### 2.1. The Circular Economy Concept—Need for Change?

The current functioning of the linear economic model has contributed to the deterioration of the natural environment. The linear economy transformed the Earth into a world in which all natural resources were harvested, processed and then delivered as products to customers. After use, they were subjected to disposal processes. Mass consumption and production have created a threat not only to the planet's resilience but also to the future of humanity [14]. Excessive use of natural resources and the creation of huge amounts of waste have become the cause of environmental problems, i.e., decline in biodiversity, soil degradation, eutrophication, release of greenhouse gases, hydrologic changes and the negative impact of artificial fertilizers [15]. Anthropogenic activities have also contributed to an increase in global earth temperature by approximately 1.0 °C compared to pre-industrial levels. It is estimated that this temperature will increase to 1.5 °C by 2052 while maintaining current emission levels. Climate change has led to natural disasters that have affected approximately 68.5 million people and economic losses of approximately USD 130 bil-

lion [16]. Climate changes caused by human activity have contributed to changes related to, among other things, biodiversity. Within ten years between 2006 and 2015, the populations of fish, mollusks and reptiles experienced a rapid loss of biodiversity: 8.0%, 8.5% and 12.5% per year [17]. Moreover, the global biodiversity status indicator in 2020 indicated a loss of plant and animal populations by approximately 68% over approximately 30 years (between 1970 and 2016) [18]. A significant environmental problem is also the huge amounts of pollutants generated in the form of solid waste, atmospheric and water pollution. According to Yorko and Daramola [19], in 2019 the average concentration of carbon dioxide in the atmosphere was 45% higher compared to the period between 1980 and 1990. Moreover, global $CO_2$ emissions caused by human activity since 1950 have increased by almost 400%. To counteract environmental problems, several concepts have been created, e.g., Sustainable Development. An important change that can help reduce the negative impact of human activity on the natural environment is following the assumptions of the circular economy.

The CE concept is an economic system which concentrates on, e.g., reduction, materials recovery and recycling. It can be realized on all three economic levels: micro, macro and meso [20]. The CE concept also applies to regeneration, closing loops, sharing and exchange. Geissdoerfer et al. [21] draw attention to the importance of the regenerative nature of the concept. They point out that the CE concept focuses on minimizing the use of resources and energy, as well as reducing the generation of waste. This can be achieved by slowing down, closing and narrowing the circulation of materials and energy. The CE concept concentrate also on the minimalization of raw materials use and energy [22]. The concept of CE focuses, in the simplest way, on three strategies, recovery, recycling and reuse, the use of which contributes to extracting value from waste and improving efficiency in relation to sustainable development. Moreover, the product life cycle changes under the influence of maximizing the use of resources and generating value in the final phase of use [23]. An important assumption of the CE concept is to strive to achieve the goals of the Sustainable Development concept [24]. The first two sustainable development goals indicate the importance of eliminating poverty and hunger, but also ensuring adequate food security and better nutrition [25]. To be effective, the circular economy concept should cover the entire production process, including using sustainable raw materials and energy sources, designing efficient production processes, developing durable and repairable products, reusing and recycling and incorporating circular business models [26]. The circular economy concept can also be defined using the 10R strategy. Unlike the 3R model, the strategies within the 10R model are better expanded and present activities that result in loss of value. They are listed from the least circular to the most circular and include recover, recycle, repurpose, remanufacture, refurbish, repair, reuse, reduce, rethink, refuse [27]. It is possible to implement the circular economy concept using individual strategies. In the literature, it is possible to describe the circular economy concept through the possibility of regeneration [28,29]. Others, in turn, pointing to the definitions of the circular economy concept, draw attention to the aspect of the possibility of closing the loop [30,31]. Analyzing the concept, we can point to a diverse approach to the concept of the circular economy. As can be seen, the literature indicates that there are various possibilities of proceeding in accordance with the assumptions of the circular economy concept. Appropriate use of strategies such as recycling or recovery can improve the environmental performance of the organization and thus contribute to the implementation of the assumptions of the circular economy concept.

The concept of the circular economy is now a global phenomenon that has covered virtually the entire planet. In many European countries, but also in the United States and Japan, activities are being implemented to develop circular economy models [32,33]. The use of various practices related to the circular economy concept also affects the functioning of individual countries. However, the perception of the circular economy concept and its implementation vary. Research conducted in Asia on the acceptance of practices related to the circular economy concept indicates that local consumers are reluctant to accept recycled or regenerated products because they value high quality and reliability of products.

Interestingly, however, they willingly use sharing platforms [34]. Comparing the circular economy concept in China with the European Union, McDowall et al. indicate that in both cases the circular economy concept has a common conceptual basis and point to similar concerns related to achieving resource efficiency. However, in the case of China, the concept of the circular economy is related to the aspect of pollution and sustainable development, while in Europe there is a focus on waste and opportunities for industry [35]. In turn, the implementation of the circular economy concept in Latin America is associated with increased interest in this topic. However, cultural and political issues that Europe is gradually struggling with turn out to be problematic in this area. However, in Latin American countries, the focus on the circular economy concept is primarily about the ability to generate economic value using recycling [36]. There are also relationships between aspects related to the concept of the circular economy, i.e., generation of municipal waste, and aspects including economic. According to research conducted by Apostu et. al., gross fixed capital formation is a factor limiting the amount of waste generated, and GDP, in turn, is the opposite. Similar relationships in the case of GDP are also observed with energy consumption and SOx and NOx emissions [37]. As can be seen, there is variation in the perception of the circular economy concept in different regions of the world. It results primarily from differences related to economic development. Latin American countries focus primarily on recycling. However, in European countries and China, attention is paid to more circular solutions related to reducing the consumption of natural resources.

Currently, the CE concept has become one of the most important development models that would reduce the impact of human activity on the natural environment. Acting in accordance with its principles or assumptions is undoubtedly necessary due to the existing environmental problems. Introducing changes towards the circular economy concept should start from the lowest economic levels, so it is important to check whether the micro level sees the need to act in accordance with its assumptions. Therefore, a research hypothesis was formulated:

**H$_1$.** *More than half of the representatives at an above-average level of management express a positive opinion on the need to act in accordance with the principles of the circular economy concept in organizations.*

### 2.2. The Circular Economy in Organizations—Management Systems and Organization Size Perspective

It is possible to implement the assumptions of the circular economy concept indirectly using management systems standards (MSSs). The key system that can significantly support the implementation of the assumptions of the circular economy concept is the environmental management system (EMS), which is compliant with the requirements of the ISO 14001 standard. The implementation of an EMS improves the environmental performance of the organization by implementing better technologies, which include, among others, the protection of natural resources [38]. Moreover, the implementation and certification of an EMS in accordance with the ISO 14001 standard contributes to reducing the amount of waste generated [39]. ISO 14001 contains requirements whose application in an organization allows for minimizing the negative impact on the environment by reducing pollutants and emissions [40]. The same applies to the circular economy concept. According to research conducted by Kumar et al. [41], one of the most important benefits resulting from the implementation of the circular economy concept may be the reduction in the amount of generated waste. Like the EMS, the circular economy concept aims to increase production efficiency and reduce the amount of natural resources used [27]. Other management systems are also important for the circular economy concept, i.e., an energy management system consistent with the requirements of the ISO 50001 standard for food safety management systems, e.g., ISO 22000.

Implementing a management system in accordance with the requirements of the ISO 50001 standard contributes to increasing and developing the organization's energy efficiency [42] and, at the same time, reducing its consumption and greenhouse gas emis-

sions [43]. The ISO 50001 standard focuses on energy consumption but also takes into account the possibility of using renewable energy generated in the system developed on its basis [44]. The circular economy concept also seeks to increase the use of renewable energy sources, which may be more effective sources [45]. Apart from improving the efficiency of resource use, one of the goals of the circular economy concept is also to make more efficient use of energy and reduce greenhouse gas emissions [46].

In turn, food safety management systems, e.g., those compliant with the requirements of the ISO 22000 standard, strive to reduce the number of errors and, simultaneously, reduce food losses in entire food supply chains [47]. Similarly, the circular economy concept seeks to minimize waste [48].

It is possible to use MSSs to implement the circular economy concept in organizations in different areas. Therefore, the aspect that becomes interesting is not only the connections between the circular economy and management systems but also determining whether the level of involvement in managing the organization may be important for the very need to act following the principles of the circular economy in organizations. Therefore, a research hypothesis was formulated:

**H$_2$.** *The higher the management level in an organization, the higher the need to act in accordance with the principles of the circular economy concept.*

The implementation of various initiatives in organizations may depend on many factors, including the size of the organization. For example, according to Bravi et al. [49], large and medium-sized companies are more open to the aspect of management system certification compared to smaller organizations or micro-enterprises. On the other hand, implementing assumptions related to environmental management, looking through the prism of sustainable development, becomes a desirable tool in fulfilling environmental responsibility by all groups of organizations [50]. However, implementing an EMS for smaller organizations is less beneficial than in the case of implementing the system in large and medium-sized organizations [51]. As one can see, there are differences in the implementation of initiatives such as management systems in organizations, taking into account their size. However, unlike the circular economy concept, management systems are better known and have appropriate guidelines for their implementation. Therefore, implementing the circular economy concept may cause barriers for every organization, regardless of its size.

The concept of the circular economy is a completely different approach to managing an organization. However, applying the assumptions of the circular economy concept is not always possible. According to research prepared by Ormazabal et al. [52], the implementation of the circular economy concept is a challenge for small and medium-sized organizations, especially when they operate in B2B relationships, as well as for producers of perishable products, such as food. They cannot control what happens to the final product. Due to the loss of control, they are unable to recover materials and are limited only to taking actions in the field of cleaner production practices in the organization. The implementation of circular economy initiatives is problematic not only for the small and medium-sized enterprise sector. Large organizations also struggle with many barriers. The size of the organization may influence secondary factors, such as technology development or access to resources, on which the implementation of the circular economy concept depends [53]. The aspect of formulating guidelines and regulations for the CE is also becoming an important issue. This aspect is more suitable for large organizations than for small and medium-sized enterprises. Bureaucratic issues in administration are also becoming a problem, reducing the possibilities of implementing the CE concept in small and medium-sized enterprises [54,55].

Implementing the circular economy concept in an organization turns out to be problematic for all sizes of organizations. For this reason, the size of the organization may not be important in the perception of the need for its implementation in the organization. Therefore, a research hypothesis was formulated:

**H₃.** *The size of the organization does not matter when assessing the need to follow the principles of the circular economy concept.*

### 2.3. The Food Industry and the Circular Economy

Food waste is one of the main problems the world faces. Currently, approximately 800 million people suffer from hunger and two billion suffer from food shortages. The reasons for this state are poverty and the lack of developed food systems. Food production is one of the main sectors contributing to greenhouse gas emissions. It also uses significant water resources and large land areas. One-third of global food production is lost or wasted throughout the food chain, from production to consumption. Food waste contributes to both an increase in demand for food production, and also has a negative impact on the environment [56]. According to Lopez-Barrera and Hertel [57], the amount of resources used for food production that are wasted constitutes one-quarter of the global use of arable land and fertilizers. In the EU alone, 88 million tonnes each year of food waste is generated throughout the supply chain [58]. As population and prosperity increase, demand for food, feed and energy will impact natural resources [59]. Therefore, food production and water consumption are expected to increase by approximately 60 and 50% by 2050. Food systems alone use approximately 30% of total energy consumption and approximately 70% of global freshwater consumption. Therefore, the current state of exploitation of natural resources needed to meet the nutritional needs of humanity is becoming one of the main causes of environmental degradation and threatens food security in the long term [60]. Food waste also translates into the waste of resources used in their production and distribution, i.e., water, fuels, fertilizers and raw materials. The effect of waste is not only on the environment in the form of resource use or degradation of ecosystems, but also affects the financial sphere and people's health. The entire food supply chain accounts for 31% of greenhouse gas emissions and 50% of eutrophication. In turn, the carbon footprint for food waste at all stages of the supply chain is 4.4 Gt of $CO_2$ [61]. The level of food losses from production and processing in Europe in 2016 was 26 million tonnes. Food waste alone accounted for 61 million tonnes, of which approximately 33% came from the wholesale, catering and retail sectors. The remaining waste came from households [62]. It is therefore crucial to reduce food losses and waste throughout the supply chain. This will both contribute to an increase in the efficiency of the use of natural resources, but may also translate into a decrease in the environmental burden caused by this sector [63]. The level of food waste depends on the wealth of the country. Countries with low income levels experience the highest levels of waste at the production, storage and processing stages. In this aspect, it is mainly a consequence of technical and managerial limitations. In contrast, in high- and middle-income countries, waste occurs at the stages of distribution and consumption [64].

One of the main aims of this CE concept is to reduce waste [65]. The CE concept can be implemented in two cycles in economic systems: technical and biological. The biological cycle refers to renewable materials that have been designed to return to the biosphere and are collected in the form of a cascading resource cycle. The phases within this cycle are intended to maintain, among others, quality of resources and waste hierarchy. Biological ingredients are called materials or products that are designed to be returned to the biosphere through degradation by microorganisms or as food for animals [66]. Food recovery resulting from the use of biological cycles may enable a reduction in food waste, limiting excessive amounts of production and redistributing food products in the supply chain [67]. According to Ouro-Salim and Guarnieri [68], solutions resulting from the circular economy concept that may allow for reducing food losses and waste can be found through reuse, recovery, closing cycles, composting food waste, re-use for animal feed, production of biomaterials, etc. Attempts to reduce food waste are already visible, among others, in the UK. In this country, the food industry has agreed to reduce waste by 20% between 2015 and 2025. The UK also plans to achieve zero greenhouse gas emissions from the food industry by 2050 [69]. The concept of a circular economy is also related to the agri-food sector in the

context of sustainable development. Sustainable development is the basic assumption of the circular economy concept, which consists of ecological, social and economic components. In the literature, there are inverse relationships between economic and environmental sustainable development [70,71]. This situation occurs in, among others, EU countries that were characterized by moderate durability of the agri-food system. The implementation of ecological assumptions in organizations of the agri-food sector must be gradual due to these differences. This is necessary because the exclusion of SPLs that have a significant impact on the socio-economic development of the EU may have a huge negative impact and encourage depopulation [71].

Considering the CE concept in the food industry may also contribute to minimizing climate impacts. According to a study carried out in Spain on the life cycle assessment of canned tuna, it was determined that the combination of the tuna canning process with the valorization of bio-waste and the production of tuna pâté reduces the environmental impact by approximately 0.03 kg of carbon dioxide equivalent per can of tuna [72,73]. Another possibility of implementing the assumptions of the circular economy concept in food processing is the use of non-thermal processing methods, such as microfiltration during pre-treatment or particle separation. Applications include thermal processing of food which contributes to the degradation of food, making it unsuitable for consumption and becoming waste. Food waste through non-determined processing methods is less harmful because it can allow for food recovery. It also contributes to ensuring adequate food safety. Conventional thermal methods ensure microbiological stability but are unable to ensure chemical safety because they generate the production of dangerous chemicals, such as polycyclic aromatic hydrocarbons [74].

Food waste is a key problem nowadays. It contributes to huge losses of food and, consequently, raw materials for its production, energy (including fuels) and water. Reducing this waste may contribute to reducing the negative impact of this sector on the natural environment. Achieving this goal is possible by using solutions of the circular economy concept. However, do organizations operating in the food industry sector see the need to use the circular economy concept in organizations? Therefore, research was carried out to determine the opinion of organizations regarding the need to take actions aimed at acting in accordance with the principles of the circular economy concept. Since the food industry has a huge impact on the natural environment, there is a need to determine whether they are more aware of the need to act in accordance with the principles of the circular economy concept. The research hypothesis was formulated:

**H$_4$.** *Representatives of organizations with an implemented FSMS perceive a greater need to act in accordance with the principles of the circular economy concept in the organization than representatives of organizations in which FSMSs are not implemented.*

### 3. Methodology

*3.1. Aim*

The aim of the study is to check the opinions of representatives of organizations on the need to act in accordance with the principles of the CE concept in organizations and to indicate the differences in these opinions. Differences between groups were assessed in three categories: the number of implemented management systems, the FSMS implementation and the size of the organization in terms of the number of employees. The research sample consisted of organizations that had at least one management system implemented, i.e., a quality management system consistent with the requirements of the ISO 9001 standard, an environmental management system consistent with the requirements of the ISO 14001 standard or food safety management systems (BRC, IFS, ISO 22000).

*3.2. Methods*

The research presented in the article is a fragment of broader research, which also covered other issues related to the implementation of the circular economy concept in organizations, but they will not be presented as part of this publication. The entire survey

questionnaire consisted of a total of 12 questions. This publication used only part of one matrix question that concerned perceptions of the CE concept in the organization. This question was developed using a Likert scale (1—definitely not, 2—probably not, 3—I don't know/no opinion, 4—probably yes, 5—definitely yes). The analysis also used a demographic question regarding the size of the organization.

The entire survey questionnaire was developed based on a literature review and a pilot study using the case study method (Figure 1). After developing the questionnaire, it was subjected to pilot testing to check the comprehensibility and readability of the survey (conducted among 10 representatives of organizations with implemented management systems).

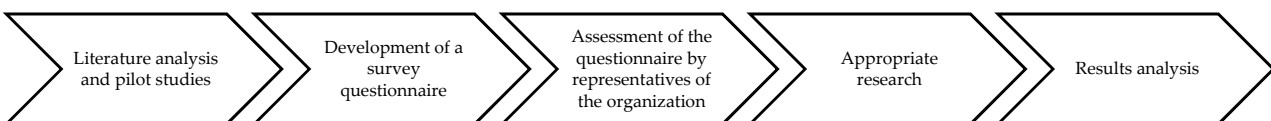

**Figure 1.** Research process.

The research was conducted using a survey questionnaire using two methods: computer-assisted telephone interview (CATI) and computer-assisted web interview (CAWI). Two data collection methods were used due to the fact that some of the research (CATI) was carried out by an external entity. Research using the CATI method was conducted in June 2023. However, research using the CAWI method was conducted from June to August 2023. In the case of research conducted using the CAWI method, the period of collecting surveys was longer due to the fact that they were conducted by the authors of the publication and we wanted to obtain as many responses as possible. Research conducted using the CAWI method took longer due to the low return of surveys.

The use of both survey methods allowed for collecting a total of 132 surveys, which were verified. The purpose of verifying the completed surveys was to reject those questionnaires that were completed by representatives of organizations that did not have implemented management systems. Based on the verification, it was found that there was no management system implemented in one organization. It was therefore rejected from further analysis. As part of the analysis of the results, the surveyed organizations were divided into groups taking into account four factors: the type of opinion, the total number of implemented management systems, the implemented food safety management systems and the size of the organization.

The level of management was measured by the number of management systems implemented in the organization. The respondents were asked about the implemented and certified management systems in the organizations. The most popular MSSs among the respondents were systems described in such standards as ISO 9001, ISO 14001 and ISO 45001. For the purpose of this study, each management system certified by independent certification body was calculated as one. The need to act in accordance with the principles of the circular economy concept was determined on a 5-point Likert scale in the question. The question asked was: "Please indicate to what extent the implementation of the circular economy is needed in your organization? Please assess in 5-point scale, where 1 is the smallest need and 5 is the highest need".

The research used a survey questionnaire and the CATI and CAWI methods, which are one of the most popular tools for conducting research in order to obtain a larger number of answers. As shown by the purpose of the research, the key to the data analysis was differences in the perception of the need to act in accordance with the assumptions of the circular economy concept among organizations. For this reason, in order to analyze the collected questionnaire data, statistical tools were used to assess the significance of differences in groups.

*3.3. Statistical Analysis Methods*

The following were used, among others: descriptive statistics or the Shapiro–Wilk normality test. The Shapiro–Wilk test is used to assess normality distribution and is a well-

established and effective test. Its use was extended by Royston to apply to sample sizes ranging from 50 to 2000 [75]. Based on the normality test, it was found that the variables were not normally distributed ($p = 0.000 < 0.05$). This is the reason the non-parametric Mann–Whitney U test was used to analyze the data between groups (samples). According to Rahardja et al. [76], the Mann–Whitney test is the most popular non-parametric test used to compare two groups. This test was used in other studies in which the variables were not normally distributed [77,78], and the scale was ordinal [79]. For the Mann–Whitney U test, the alternative hypothesis, meaning that there is a difference (with respect to the central tendency) between the two groups (samples) in the population, is valid when the $p$ value is 0.05 or smaller. For all calculations, the studied groups were larger than 25 cases.

*3.4. Characteristics of the Research Sample*

In the conducted research 132 responses were collected. One of them was rejected and ultimately 131 responses were collected. The respondents in the research were representatives of organizations operating in Poland in which various management systems were implemented, i.e., a quality management system consistent with the requirements of the ISO 9001 standard. Most often, the respondents were people responsible for management systems in their organizations. The surveyed respondents were representatives of organizations that were diverse in terms of their type of activity. The characteristics of the organization sample, taking into account the main type of activity, are presented in Figure 2.

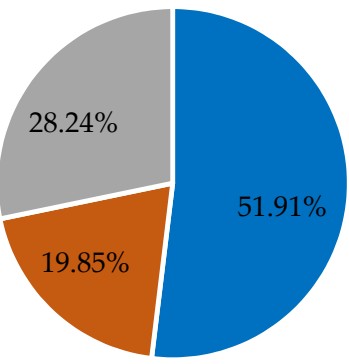

■ production  ■ services (include administration)  ■ production-services

**Figure 2.** Division of the organizations taking into account the main type of activity.

According to Figure 2, the majority of respondents were representatives of organizations dealing with production. The smallest number were purely service organizations—also administration. In each organization, at least one MSS was implemented.

The organizations studied were also characterized by diversity in terms of organization size. Figure 3 shows the percentage share of individual groups, taking into account the division into the size of the organization measured by the number of employees.

According to Figure 3, most of the surveyed organizations employ 250 or fewer employees (70.23% of the organizations). The rest of the organizations (29.77%) employ more than 250 employees.

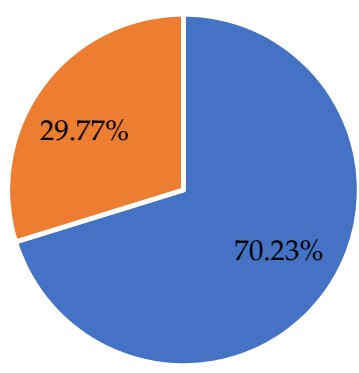

● 250 or less employees   ● More than 250 employees

**Figure 3.** Division of the organizations taking into account the size of the organization.

## 4. Results

### *4.1. Descriptive Analysis*

In the first part, a descriptive analysis was performed taking into account four criteria: the type of opinion on the need to implement the circular economy concept, the size of the organization, level of management (the number of implemented management systems) and having an implemented (at least one) FSMS. The results of the analysis are presented in Table 1.

**Table 1.** Descriptive analysis.

| Criterium | Group | Number of Organization | Percentage Share of the Sample [%] | Mean in Groups |
|---|---|---|---|---|
| Type of opinion | Positive opinion (group 1) | 98 | 74.8 | 4.29 |
|  | Negative or neutral opinion (group 2) | 33 | 25.2 | 2.79 |
| Level of management | Three or fewer MSSs (group 1) | 95 | 72.5 | 3.79 |
|  | More than three MSSs (group 2) | 36 | 27.5 | 4.22 |
| FSMS implementation | No FSMS (group 1) | 103 | 78.6 | 3.88 |
|  | At least one FSMS (group 2) | 28 | 21.4 | 4.00 |
| Organization size (in number of employees) | 250 or less employees (group 1) | 92 | 70.2 | 3.89 |
|  | More than 250 employees (group 2) | 39 | 29.8 | 3.95 |

Notes: Prepared based on own research.

According to data in Table 1, it can be concluded that the groups are characterized by uneven distribution. Typically, the first group consists of three times as many organizations as the second. In the case of the first two criteria, the groups have different average responses. However, in the case of the remaining two criteria, the averages in the groups are similar.

## 4.2. Analysis of the Differences between Groups

In the second part of the study, the significance of differences in groups was analyzed using the Mann–Whitney U test. The results of the analysis are presented collectively in Table 2.

**Table 2.** U Mann–Whitney test results.

| Criterium | Rank Sum (Group 1) | Rank Sum (Group 2) | U | Z | p | Z cor. | p |
|---|---|---|---|---|---|---|---|
| Type of opinion | 8085 | 561 | 0.00 | 8.571 | 0.000 | 9.419 | 0.000 |
| Level of management | 5788 | 2858 | 1228 | 2.482 | 0.013 | 2.728 | 0.006 |
| FSMS implementation | 6728 | 1918 | 1372 | −0.390 | 0.696 | −0.429 | 0.668 |
| Organization size | 2674 | 5972 | 1694 | 0.501 | 0.616 | 0.550 | 0.582 |

Notes: Prepared based on own research.

The first criterion for which the significance of differences in groups was checked is the type of opinion. With a significance level of $p = 0.05$, in this case the null hypothesis indicating that the groups are the same should be rejected. Therefore, there are statistically significant differences between these groups. By assessing the distribution of the variable and the average indications in both groups and the results of the Mann–Whitney U test, the assumed research hypothesis ($H_1$) is confirmed, stating that the majority of the representatives express a positive opinion on the need to act in accordance with the principles of the circular economy concept in organizations.

In the case of the second criterion, a similar situation occurs. At a significance level of 0.05 in the case of the management level criterion, the null hypothesis that the groups are similar should be rejected and the alternative hypothesis that the groups are different should be accepted. That implies that the research hypothesis ($H_2$) is confirmed. The theory indicated that the higher the number of MSSs, the greater the need to act in accordance with the principles of the circular economy concept. The hypothesis is confirmed by statistically significant differences and the mean value, which is higher in the group with more MSSs implemented than in the case of the first group. The Spearman's rank correlation for the studied variables was $R = 0.17$ with $p = 0.051$. That result does not allow us to confirm the hypothesis $H_2$.

Similar to the above, organizations were divided into groups for one, two, and four MSSs in a group. The results did not indicate statistically significant differences in the CE assessment for these groups.

In the case of the remaining two criteria, due to the fact that the *p*-value is higher than the assumed level of significance, there are no grounds to reject the null hypothesis that the groups are the same. The division of organizations according to the last two criteria showed that between these groups in their area there are no significant differences in the perception of the need to implement activities related to the concept of a circular economy in organizations. Moreover, the group means within the criteria are very close to each other. It can therefore be concluded that there are no differences between the perception of the need to act in accordance with the assumptions of the circular economy concept in organizations with and without implemented FSMSs and in larger and smaller organizations ($H_3$ and $H_4$).

## 5. Discussion

Implementing the circular economy concept is nowadays very important because it allows organizations to minimize the level of waste and limit the negative impact on the natural environment. However, a complete transition to a circular model may prove problematic for organizations. Among the barriers to implementing the concept in organizations, the literature draws attention to the need for more data, the problem of financing circular business investments, difficulties in cooperation with other organizations and high investment costs [80]. Despite these barriers, the research showed that there is a need to

act in organizations in accordance with the principles of the circular economy concept. Implementing initiatives related to the circular economy can bring results. According to research conducted by Barreiro-Gen and Lozano [81], for the surveyed organizations, good results were confirmed for 12.7% of the organizations, and certain results for approximately 33% of the surveyed organizations.

The high level of organizations that declare the need to act in accordance with CE principles is not surprising. Nevertheless, it is just a declaration of management, not proof of real engagement. The gap between declarations and real actions was indicated by the researchers. According to Rhee and Lee's [82] study, there are actually gaps between the rhetoric and reality of environmental strategy of organizations. As rhetoric changes faster than reality, the high level of declarations considering the CE principles obtained in the study can be a promising result. However, that positive scenario will depend on the attitude of managers, both at the middle and top management levels, as Faraz et al.'s study confirms a positive mechanism between leaders and employees within environmental behaviors [83].

The correlation between the number of MSSs and the CE assessment is above zero. That means that the increase in the number of MSSs is associated with an increase in declarations of the need to implement the CE. The number of implemented MSSs can be seen as an indicator of the level of management in the organization. The significant difference in our study was found within the organizations with one to three MSSs and those with more than three MSSs. One explanation of the result could be the popularity of individual MSSs in organizations. There are three most popular standards such as ISO 9001, ISO 14001 and ISO 45001 [84]. According to Ramos et al. [85], there is a tendency to implement MSSs covering the ISO 45001, ISO 14001 and ISO 9001 standards, where integration can render the process more efficient and effective, reducing bureaucracy and saving money. Those systems are well known and available on the certification market for a long time. It can be assumed that what differentiates organizations are other implemented generic management systems and industry-specific MSSs. The other possible explanation for the results concerning the number of MSSs and CE assessment can be the greater openness to other systems and requirements, including the CE, in the case of organizations that have already implemented several MSSs. As the research of Kafel and Casadesus shows, implementing subsequent management systems is easier than implementing the first one [86]. This may also involve greater openness to the goals of other systems and motivation to achieve them.

From the institutional theory perspective, the leaders and early adopters of new certification schemes achieve more significant benefits than late adopters or followers. As Yang et al.'s [87] study shows, on the example of OHSAS implementation, that theory is valid for the occupational health MSS. It is possible that organizations with more than three MSSs can be considered leaders in MSS implementation nowadays. Such an organization could perceive the need to implement CE principles as an opportunity to achieve benefits similar to previously implemented systems.

The other important aspect that should be discussed is the calculation of the number of MSSs. In this study, even very similar standards, such as BRC and IFS, were calculated separately. The same was done for ISO 14001 and EMAS. These systems are certified independently, but the main goal, requirements and implementation can be quite similar [88]. Due to that, there is a need to further investigate what further differentiates the approach to assessing the CE. A greater number of systems or a variety of goals that are achieved thanks to these systems.

The conducted research did not indicate significant statistical differences in the assessment of the importance of CE implementation, considering food systems as a distinction. The food industry is characterized by a significant impact on the natural environment, mainly because this is where huge product losses and waste occur. This also translates into wastage of energy, fuels, fertilizers, and consequently the raw materials themselves. This adversely affects the functioning of ecosystems and human health and is also reflected in

economic aspects. Therefore, it becomes important to look for solutions to this problem. The circular economy concept, which sets this aspect as one of its goals, may be crucial in the fight against waste. Implementing the circular economy concept in the food industry may allow for a reduction in waste, thus reducing the industry's negative impact on the natural environment. As research related to the motives for implementing environmental systems shows, organizations with a greater negative impact on the environment indicated greater involvement in such activities. Moreover, ISO 14001 certification raises awareness of compliance requirements with relevant legislation [40]. Similar results are indicated in the case of the OHSAS system [87]. Similarly, organizations in the food sector should show greater interest in implementing CE principles. In line with that reasoning, the lack of statistical verification of the $H_3$ hypothesis may be caused by the participation of small and medium-sized organizations in the study group. Such organizations generally have a lower impact on the environment. Due to that, there is a need to investigate that topic further.

The conducted research indicates that the development related to organizational management translates into the perception of the circular economy concept by organizations. In the case of organizations that have already implemented management systems, their improvement and strengthening in the form of implementing further organizational management tools may translate into greater opportunities for assimilating the circular economy concept in the organization. It turns out that not only environmental management systems are an important element of the implementation of the circular economy concept. It is therefore crucial to pay attention to aspects related to the development of the organization in terms of management, which may facilitate and allow for a greater understanding of the principles behind the concept of the circular economy. The owners of the organization may not be fully aware that the implementation of a given management system also brings environmental benefits that result from the assumptions of the circular economy concept, such as an FSMS which limits food waste. These studies did not show any relationship between having an implemented FSMS and the need to act in accordance with the principles of the circular economy concept. Nevertheless, a key limitation here is the number of organizations analyzed in the study. To better assess this aspect, it would be necessary to conduct such research on a larger number of organizations that have implemented FSMSs or to conduct it in other countries. This criterion could have different results in a larger number of respondents.

## 6. Conclusions

Following the principles of the circular economy concept is extremely important but a big challenge for organizations, not only in the food industry. The purpose of this article is to determine the perception of the need to act in accordance with the principles of the CE concept in organizations operating in Poland and having implemented management system standards. Based on the study, it was found that 75% of the surveyed organizations see the need to act in accordance with the assumptions of the circular economy concept. Interestingly, almost every organization that represented the food industry expressed only positive opinions on this aspect. It can, therefore, be concluded that the surveyed respondents from the food sector notice problematic aspects related to the functioning of this industry and see the need for a change towards a closed loop. An analysis of the significance of differences between groups was also performed. When dividing organizations according to the number of implemented MSSs, it was found that there are statistically significant differences between groups with the number of MSSs three or fewer and more than three in the need to act according to the rules CE. The organizations with more than three MSSs perceive a greater need to act in accordance with the principles of the CE than other companies. The division taking into account the size of the organization showed that this factor is not important when it comes to perceiving the need to act in accordance with the principles of the circular economy concept. In the opinion of representatives of organizations with a larger number of management systems, acting in accordance with the assumptions of the circular economy concept is necessary in the organization. However,

it becomes problematic to understand what these representatives believe the concept of the circular economy is. The literature indicates various approaches to understanding the circular economy concept, which can be implemented using the least circular strategies, e.g., recycling.

The study has some limitations that should be expressed. Firstly, the study sample included only organizations with at least one implemented and certified management system. Such organizations are generally characterized by a higher level of management than organizations that have not implemented this type of management system. The other limitation is that the research is limited only to one country with an average level of implementation and CE requirements. The other limitation of the study is the range of analysis performed within this study. As the circular economy is a complex subject and the data used only concerned recipients' CE perception, there is still a need for further in-depth analysis.

The research shows that the level of involvement in managing the organization is important when assessing the need to act in accordance with the assumptions of the circular economy concept. This study only checked whether FSMSs have a significant impact in this aspect. Therefore, it would be necessary to develop research and check how opinions on acting in accordance with the assumptions of the circular economy concept are distributed in the case of other management systems individually. Perhaps there are some specific management systems responsible for the differences between the level of involvement in managing the organization and the need to act in accordance with the principles of the circular economy concept. These studies only take into account organizations that have implemented management systems. However, not all organizations operating in a given country have implemented management systems. Therefore, another development of the research could be to determine opinions on the need to act in accordance with the assumptions of the circular economy concept in organizations that do not have these systems implemented. An interesting development of the research could be conducting research on customers. As we know, they are one of the driving forces of every organization and the demand for a given organization's products depends on them.

The research conducted concerned only organizations operating in Poland. However, we believe that it is possible to conduct similar research in other countries, such as Germany or the Czech Republic, and to compare opinions on the need to act in accordance with the assumptions of the circular economy concept internationally. As an important extension of the research, it would be possible to check the opinion on the rules of conduct in accordance with the assumptions of the circular economy concept in organizations that do not have any management systems. Currently, most small businesses do not implement this type of initiative. Nevertheless, they may take actions aimed at implementing the circular economy concept and, therefore, may feel the need to act in accordance with its assumptions. For this research, the focus was on the food industry. However, we believe that there is a need to check the opinions of other types of industry, e.g., clothing, which also generate huge amounts of waste and consume raw materials.

**Author Contributions:** Conceptualization, P.K.; Formal analysis, K.R.; Writing—original draft, K.R.; Supervision, P.K. All authors have read and agreed to the published version of the manuscript.

**Funding:** The publication/article presents the result of the Project no 061/ZJJ/2023/POT financed from the subsidy granted to the Krakow University.

**Institutional Review Board Statement:** The study was conducted in accordance with the Declaration of Helsinki, and approved by the Institutional Review Board of Krakow University of Economics.

**Informed Consent Statement:** Informed consent was obtained from all subjects involved in the study.

**Data Availability Statement:** Data are contained within the article.

**Conflicts of Interest:** The authors declare no conflicts of interest.

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
