# Peer review of "The Importance of the Circular Economy Concept among Organizations within the Food Sector and a Management Systems Perspective"

_sustainability, doi:10.3390/su16072912_

Round 1

Reviewer 1 Report (Previous Reviewer 1)

Comments and Suggestions for Authors

Once again I would like to thank the Editorial Board of Sustainability for the opportunity they have given me to serve as a reviewer for this prestigious publication. Now focused on the review of the paper entitled: “The importance of circular economy concept among organizations within the food sector and management systems perspective" (Manuscript ID sustainability-2919032), below, I indicate what has been my decision in this regard:

Accept in present form

I have been able to verify that the authors conscientiously followed the indications, suggestions, and proposals for improvement that I gave them previously, so that this paper has substantially improved its academic quality, being worthy of being published in this publication. Congratulations.

With my best wishes in your personal and academic life,

The reviewer

Author Response

Thank you for your review.

Reviewer 2 Report (New Reviewer)

Comments and Suggestions for Authors

The authors have produced an article of interest. In the article entitled "The importance of circular economy concept among organizations within the food sector and management systems perspective", the authors identify the opinion of the sector representatives to act in accordance with the principles of circular economy. However, the authors need to make some changes before considering the article for publication.

Introduction

It is solid, the authors make an adequate exposition of the gaps.

Literature review

The authors have conducted an adequate literature review. Although they should exploit other points of view not indicated. For example, Kirchherr et al. (2017) indicate that the concept of circular economy has undergone a process of redefinition. In that process, the authors indicate that, in some publications, they have tried to "eliminate" the principle of reducing from the concept of circular economy, focusing it only on the use of by-products and the recycling of waste, so that companies only have to improve the recycling rate and the rate of use of by-products. Please indicate such reflection.

The authors should relate more the circular economy to the sustainability of the agri-food sector. I suggest two publications:

Castillo-Díaz, F. J., Belmonte-Ureña, L. J., Batlles-delaFuente, A., & Camacho-Ferre, F. (2023). Strategic evaluation of the sustainability of the Spanish primary sector within the framework of the circular economy. Sustainable Development. https://doi.org/10.1002/sd.2837

Castillo-Díaz, F. J., Belmonte-Ureña, L. J., López-Serrano, M. J., & Camacho-Ferre, F. (2023). Assessment of the sustainability of the European agri-food sector in the context of the circular economy. Sustainable Production and Consumption, 40, 398–411. https://doi.org/10.1016/j.spc.2023.07.010

Methodology

Authors should include an infographic summarizing the research process.

Authors should differentiate the statistical treatment section, it is a bit messy to be found in a general section of Aim, methods and hypotheses.

Results

The presentation and comments on the results is adequate.

Discussion

The authors should improve the discussion. The authors do not make recommendations for policy makers.

Conclusion

The authors should expand on the theoretical and practical aspects of their research.

The authors should expand the future lines of research that emerge from their research.

Author Response

First, thank the reviewer for their evaluation and valuable suggestions. We hope that the added changes are coherent with the reviewers’ requests, improving the quality of the paper.

This manuscript is a resubmission of an earlier submission. The following is a list of the peer review reports and author responses from that submission.

Round 1

Reviewer 1 Report

Comments and Suggestions for Authors

Once again I would like to thank the Editorial Board of Sustainability for the opportunity they have given me to serve as a reviewer for this prestigious publication. Now focused on the review of the paper entitled: "The importance of circular economy concept among organizations within the food sector and management systems perspective" (Manuscript ID sustainability-2832754), below, I indicate what has been my decision in this regard:

Reconsider after major revision

In my opinion, the paper is to some extent "incomplete", that is, there are some parts that should be reworked. Here are some of them:

1. At the end of the introduction, include a short paragraph summarizing the topics that will be addressed in the following sections.

2.         Indeed, the Circular Economy is a global phenomenon that today covers practically the entire planet. In this regard, please include a paragraph arguing how the practices derived from the Circular Economy affect all continents, i.e., Europe, Asia or America. In this regard, please include the following references:

  • KUAH, Adrian T.H. and WANG, Pengji. "Circular economy and consumer acceptance: An exploratory study in East and Southeast Asia". Journal of Cleaner Production. 2020, vol 247, https://doi.org/10.1016/j.jclepro.2019.119097

  • APOSTU, Simona Andreea, GIGAURI, Iza, PANAIT, Mirela and MARTÍN-CERVANTES, Pedro A.. "Is Europe on the Way to Sustainable Development? Compatibility of Green Environment, Economic Growth, and Circular Economy Issues". International Journal of Environmental Research and Public Health. 2023, vol 20, num. 2, p. 1078. https://doi.org/10.3390/ijerph20021078

  • BETANCOURT MORALES, Claudia Marcela and ZARTHA SOSSA, Jhon Wilder. "Circular economy in Latin America: A systematic literature review". Business Strategy and the Environment. 2020, vol 29, num. 6, p. 2479 – 2497. https://doi.org/10.1002/bse.2515

  • MCDOWALL, Will, GENG, Yong, HUANG, Beijia, BARTEKOVÁ, Eva, BLEISCHWITZ, Raimund, TÜRKELI, Serdar, KEMP, René and DOMÉNECH, Teresa. "Circular Economy Policies in China and Europe". Journal of Industrial Ecology. 2017, vol 21, num. 3, p. 651 – 661. https://doi.org/10.1111/jiec.12597

3. On the methodological part, could you give a more convincing explanation as to why the two types of questionnaires did not coincide in time (CAWI vs. CATI).

4.         In the methodological part, I miss that you discuss at some length which cities were covered by the surveys. Please indicate.

5.         In the methodological part the authors indicate "The results of the conducted research were analyzed using the Statistica program". Please remove this line. In my opinion, this is a somewhat outdated custom and the important thing is not to specify the computer program used, but that the results are robust and congruent. It is assumed that with another program the results would be exactly the same.

6.         In the methodological part, the authors should specify why they have used this specific methodology, what have been the specific reasons, instead of other methodologies focused on the analysis of questionnaires (i.e. neural networks).

7.         In lines 303-307 replace the paragraph with a "pie chart" type graph. I believe that your work could have gained much more if you had chosen to include a greater number of explanatory graphs (you do not include any).

8. "Too many tables" for "so little text". That said, unify tables 1-2 , 3-4 and, 5-6, in only two tables in panel form.

9.         In the conclusions, please delineate more precisely what new lines of research and/or practical policies could be pursued.

 10.       Finally, in the conclusions, include a last paragraph in which you reflect on whether it is possible (or not) to replicate this research in other countries in the immediate area where the research was carried out.

 With my best wishes in your personal and academic life.

Author Response

First, thank the reviewer for their evaluation and valuable suggestions. We hope that the added changes are coherent with the reviewers’ requests, improving the quality of the paper. All details are in the attachment.

Reviewer 2 Report

Comments and Suggestions for Authors

The manuscript aim is too much speculative and constrained in audience in is present form.

The aim of the research is to determine the perception of the need to act in accordance with the principles of the circular economy concept in organizations operating in Poland and having implemented management systems.

Method:

+ The entire survey questionnaire consisted of 12 questions: 10 closed and 2 open.

The full methodology should be disclosed for a proper methodological assessment; as it can be too simple, biased, inconsistent, etc. As is provided no proper conclusions can be drawn.

Sample:

+ A total of 131 respondents took part in the study, representing various organizations dealing with production, services, and also administration.

The provided description is insufficient for assessing the suitability of the sample.

Results:

+ The statistical significance of differences between these groups was assessed using the Mann-Whitney U test.

Post hoc analysis is wrong.

Author Response

(The authors gave the same response as above.)

Round 2

Reviewer 1 Report

Comments and Suggestions for Authors

Once again I would like to thank the Editorial Board of Sustainability for the opportunity they have given me to serve as a reviewer for this prestigious publication. Now focused on the review of the paper entitled: "The importance of circular economy concept among organizations within the food sector and management systems perspective" (Manuscript ID sustainability-2832754), below, I indicate what has been my decision in this regard:

Accept in present form

Since the authors have carried out all the changes, suggestions and proposals for improvement proposed by me, in my opinion, the work has to be accepted in its present state. However, I would like to give a final word of advice to the authors. I asked them to illustrate with a certain amount of freedom in which cities the surveys were carried out and they replied: "However, in terms of the characteristics of the research sample, it was clearly stated that they covered the entire territory of Poland". Obviously, if the survey is conducted in Poland, it covers the Polish territory, I agree with you. Please be a little more explicit in your next work. Be that as it may, congratulations on the final acceptance of your work.

With my best wishes in your personal and academic life,

The reviewer

Author Response

Dear Reviewer

Thank you for your comments and valuable insights.

We wish you all best.

Reviewer 2 Report

Comments and Suggestions for Authors

As the authors do not provide the full analysis for the 12 question questionnaire, the value and the suitability of the analysis can not properly be addressed. In fact, the statistical analysis for chosen factor is wrongly made.

As circular economy is a complex subject and proper research has to be made, refined methods and analysis need top be consistent and not to fulfill researcher's agenda.

Author Response

Thank you for your comment. We strongly agree with you, that the circular economy is a complex subject and cannot be described on the basis of one study. We have used only the question concerning the perception concerning the CE for the purpose on this article and we believe that the results add value to existing CE-related knowledge. The other questions from the study do not cower the perception of CE but are focused on the issues such as e.g. environmental and quality management systems and its relation to the CE.

Taking into account your comment, we have clarified the description of the selected questions and the limitations of the research. Please take in mind that more complex analysis with the use of all questionnaire data is currently performed as part of the phd thesis and we appreciate all the possible feedbeck. We would appreciate your suggestion and expertise considering the choose of more suitable statistical analysis that we can use in the further research.

With my best wishes in your personal and academic life.

Round 3

Reviewer 2 Report

Comments and Suggestions for Authors

No improvement on quality of analysis, statistical inference methods or full disclosure of methods are provided.

Author Response

Dear Reviewer,
Please find the new version of the text in attachment. We have added the question that was used in the study for statistical calculations. We also added the explanation how we have calculated the number management systems. We hope that this changes give a full disclosure of methods we have used to choose the groups and calculate the results.

We believe that explains all the dispels any doubts about the method of conducting statistical analysis.

Best regards

Authors